# Study on the Strength and Hydration Behavior of Sulfate-Resistant Cement in High Geothermal Environment

**DOI:** 10.3390/ma15082790

**Published:** 2022-04-11

**Authors:** Yan Wang, Yahao Chen, Bingbing Guo, Shaohui Zhang, Yueping Tong, Ditao Niu

**Affiliations:** 1College of Materials Science & Engineering, Xi’an University of Architecture and Technology, Xi’an 710055, China; chenyahao619@163.com (Y.C.); tongyueping@163.com (Y.T.); 2State Key Laboratory of Green Building in Western China, Xi’an University of Architecture and Technology, Xi’an 710055, China; guobingbing212@163.com (B.G.); niuditao@163.com (D.N.); 3College of Civil Engineering, Xi’an University of Architecture and Technology, Xi’an 710055, China; zhangshaohui999@126.com

**Keywords:** high-temperature environment, sulfate-resistant cement, pore structure, porous, hydration product, microscopic morphology, thermodynamics

## Abstract

The hydration process and compressive strength and flexural strength development of sulphate-resistant Portland cement (SRPC) curing at 20 °C, 40 °C, 50 °C, and 60 °C were studied. In addition, MIP, XRD, SEM, and a thermodynamic simulation (using Gibbs Energy Minimization Software (GEMS)) were used to study the pore structure, the types, contents, and transformations of hydration products, and the changes in the internal micro-morphology. The results indicate that, compared with normal-temperature curing (20 °C), the early compressive strength (1, 3, and 7 d) of SRPC cured at 40~60 °C increased by 10.1~57.4%, and the flexural strength increased by 1.8~21.3%. However, high-temperature curing was unfavorable for the development of compressive strength and flexural strength in the later period (28~90 d), as they were reduced by 1.5~14.6% and 1.1~25.5%, respectively. With the increase in the curing temperature and curing age, the internal pores of the SRPC changed from small pores to large pores, and the number of harmful pores (>50 nm) increased significantly. In addition, the pore structure was further coarsened after curing at 60 °C for 90 d, and the number of multiple harmful pores (>200 nm) increased by 17.9%. High-temperature curing had no effect on the types of hydration products of the SRPC but accelerated the formation rate of hydration products. The production of the hydration products C-S-H increased by 13.5%, 18.6%, and 22.8% after curing at 40, 50, and 60 °C for 3 d, respectively. The stability of ettringite (AFt) reduced under high-temperature curing, and its diffraction peak was not observed in the XRD patterns. When the curing temperature was higher than 50 °C, AFt began to transform into monosulfate, which consumed more tricalcium aluminate hydrate and inhibited the formation of “delayed ettringite”. Under high-temperature curing, the compactness of the internal microstructure of the SRPC decreased, and the distribution of hydration products was not uniform, which affected the growth in its strength during the later period.

## 1. Introduction

The “Belt and Road” strategy has led to the construction of infrastructure in western China, and the number of railways and highway tunnels has increased significantly. However, these areas are rich in geothermal resources and experience active crustal movement, which bring high-temperature geothermal problems that affect the performance of lining concrete [1]. In addition, groundwater contains an abundance of ${SO}_{4}^{2-}$ [2], which further accelerates the rate of damage to concrete structures in high-temperature geothermal environments [3,4].

In practical engineering, sulphate-resistant Portland cement (SRPC) is an effective measure for solving the sulfate attack problem of concrete [5]. The cement reduces the formation of delayed ettringite by controlling the content of aluminum phase in the clinker, thereby reducing the expansion damage to the concrete caused by the sulfate attack. The results of Aziez et al. [6] showed that after 24 months of erosion in 5% MgSO_4_ solution, the strength of SRPC mortar was 4.9% higher than that before erosion. The study of Kroviakov [7] showed that the strength of SRPC varied from 94.7 to 105.1% after immersion in 10,000 mg/L Na_2_SO_4_ solution for 1, 3, and 6 months and found that the concrete made of sulfate-resistant Portland cement had better durability. Puppala [8] also found that SPRC effectively reduces the risk of a sulfate attack on concrete by reducing the formation and hydration of ettringite. However, little research has been conducted on the erosion resistance of SRPC in a high-temperature environment.

The hydration process of cement-based materials determines the internal microstructure and various properties of cement stone, but the hydration of cement is affected by temperature [9,10]. High temperatures promote the hydration of cement, but a hydration reaction that is too fast will lead to the precipitation of hydration products on the surface of cement particles and the incomplete hydration of the cement, which will also cause an uneven distribution of hydration products [11], a decrease in the stability of the internal microstructure of the cement [11,12,13], an increase in porosity [13,14], and the decomposition of some products under high temperatures [14]. He et al. [15] found that under the conditions of high-temperature curing (temperature, 50/60/80 °C; relative humidity, 50%), the 3-day strength of concrete had obviously increased, but the 60-day strength had greatly decreased. Han et al. [16] showed that high temperatures have little effect on the final reaction degree of cement, but high temperatures reduce the non-evaporative water content of the cement paste in the later stage, which produces an uneven distribution of hydration products in the microstructure, resulting in a higher porosity and a lower high-temperature compressive strength in the later stage.

However, the amount of tricalcium aluminate (C_3_A) in SRPC clinker is small, and there are fewer aluminum-phase substances in the hydration product, which can effectively inhibit the formation of “delayed ettringite”. However, the hydration process of SRPC in a high-temperature geothermal environment is worth exploring. In addition, the composition of the cement cementitious system is complex, and transformations and losses of bound water occur under high temperatures, which further increase the complexity of the hydration behavior under high temperatures. To date, it has been difficult to fully explore the hydration process of cement by experimental means. In recent years, the thermodynamic model developed in the field of geochemical analysis, coupled with an accurate and complete thermodynamic database, has been used to reliably predict the chemical reactions between cement phases [17]. Moreover, the thermodynamic model is also an effective tool for evaluating the performance of cement in a specific environment [18]. It has been successfully applied in studies on the hydration process of materials, such as the hydration reaction of sulphoaluminate cement and fly ash [19], the hydration of an alkali slag system [20], and the hydration of ordinary Portland cement [17].

While many studies have been conducted on the mechanical properties and durability of SRPC at room temperature, the influence of high temperatures on the hydration process of SRPC, especially the development of the mechanical properties of SRPC at high temperatures, has been less researched. In this study, the strength development, the transformation of hydration product types and substances, and the evolution of the microscopic morphology and pore structure of SRPC at curing temperatures of 20 °C, 40 °C, 50 °C, and 60 °C were analyzed, and a microscopic test, a thermodynamic simulation, and a laboratory test were conducted. The research results may provide theoretical support for the application of SRPC in the engineering of tunnels under such special geological conditions and long-term performance damage evolution analysis.

## 2. Materials and Methods

### 2.1. Raw Materials

P·HSR 42.5-grade sulfate-resistant cement (Zhongde Xinya Building Materials Co., Ltd., Beijing, China) was used as the cementitious material, and ordinary Portland cement (Shaanxi Yaobo Cement Co., Ltd., Xi’an, China) was used as the control material. The chemical and mineral composition of these materials are shown in Table 1 and Table 2. The fine aggregate was medium sand with a fineness modulus of 2.6. Polycarboxylic acid superplasticizer (Jiangsu Subote New Materials Co., Ltd., Nanjing, China) was used as a water-reducing agent at a dosage of 1.0 wt.% of cementitious material.

### 2.2. Specimen Preparation

Cement mortar specimens were prepared according to the Chinese standard GB/T 17671-1999 [21]. The size of the mortar specimens was 40 mm × 40 mm × 160 mm, and the water–cement ratio was 0.4. A cement paste with the same water–cement ratio was also formed. In order to ensure the accuracy of the data, all the operation procedures in the test were carried out according to the Chinese standard GB/T 17671-1999 [21], the strength value of each specimen is the average of three groups of data, and the final test results are the average of three groups of data.

The cement mortar specimens were demolded after curing for 24 h at 20 °C and 95% relative humidity. The specimens were then cured in deionized water at 20 °C, 40 °C, 50 °C, and 60 °C to the specified age, respectively, in order to determine the compressive strength and the flexural strength of the SRPC. The cement paste was sealed in a polyethylene bottle and cured under the same conditions until it reached the specified age. Then, the specimens were crushed into 3~5 mm particles, soaked in anhydrous ethanol to terminate the hydration reaction, and dried to a constant weight at 40 °C in a vacuum-drying box for microscopic analysis.

### 2.3. Testing Method

Figure 1 shows the preparation and testing stages of the mechanical property and microscopic tests. The compressive strength and the flexural strength of the cement mortar were tested according to GB/T 17671-1999 [21]. The tested ages were 1 d, 3 d, 7 d, 28 d, 60 d, and 90 d. The XRD test used a Shimadzu XRD-6100 X-ray powder diffractometer. The 2θ scanning range was 5°~90°, and the scanning rate was 10°/min. The SEM test adopted a Hitachi S-4800 field emission scanning electron microscope with a resolution of 1 nm (1280 × 960). In order to obtain good conductivity, all samples were sprayed with gold before the test. The pore structure of the cement mortar was determined by an Autopore 9620 high-performance automatic mercury porosimeter.

The thermodynamic computational simulation was based on the Gibbs free energy minimization program GEMS [22,23] and the PSI-GEMS database [24]. The database includes thermodynamic data and some data on the minerals in cement, such as ettringite (AFt), single-phase aluminum–iron (AFm), hydrotalcite, hydrated garnet, and C-S-H. The chemical composition of cement and the conditions of the hydration reaction defined in GEMS, in combination with the Parrot–Killoh [25] empirical model, were used to obtain the relationship between the degree of cement hydration and the time. The Parrot–Killoh empirical model is given in Formulas (1)~(3).

Nucleation and growth:(1)$$R_{t}=\frac{K_{1}}{N_{1}}\left(1-\alpha_{t}\right){\left(-ln\left(1-\alpha_{t}\right)\right)}^{\left(1-N_{1}\right)}$$

Diffusion:(2)$$R_{t}=\frac{K_{2}\times {\left(1-\alpha_{t}\right)}^{2/3}}{1-{\left(1-\alpha_{t}\right)}^{1/3}}$$

Formation of the hydration phase:(3)$$R_{t}=K_{3}\times {\left(1-\alpha_{t}\right)}^{N_{3}}$$
where the degree of hydration *α* at time *t* (days) is expressed as *α_t_* = *α_t_*_−1_ + *∆t*·R*_t_*_−1_. The parameter values used in the calculations can be found in the literature [26].

## 3. Results and Discussion

### 3.1. Compressive and Flexural Strength

The development of the SRPC’s strength with age under high-temperature curing is shown in Figure 2. Compared with curing at 20 °C, the compressive and flexural strength of the SRPC at the early stage (before 7 d) were improved to some extent under high-temperature curing. The increase in the rate of development of the compressive strength is shown in Table 3. The development of a cement’s strength mainly depends on the hydration reaction of silicate minerals (the higher the reaction degree of the cementitious system, the faster the development of the cement’s strength) [27]. It can be seen from Figure 2a that high temperatures are beneficial to the development of the early strength of the SRPC. Before 7 d, the higher the curing temperature, the faster the increase in the compressive strength of the SRPC. This is because an increase in the ambient temperature accelerates the hydration reaction of cement and promotes the formation of hydration products [13]. After 28 d, the promoting effect of temperature on the development of the cement’s strength was weakened. There was no difference in the compressive strength of the SRPC under the four curing conditions at 60 d. However, at 90 d, the compressive strength of the SRPC decreased, and the flexural strength began to decrease after 28 d, indicating that high-temperature curing affected the development of the cement’s strength in the later stage. The microstructure of a cement paste affects its macroscopic properties [9,11,15]. Under high-temperature curing conditions, the internal structure of the cement paste is different from that under normal-temperature curing conditions, and the rapid early hydration reaction leads to an uneven diffusion of hydration products (see Section 3.3.2). The sparse distribution of gel components becomes the weak point of the microstructure and affects the growth of the macroscopic strength [12,13]. The test results on pore structure (see Figure 3) also provide a reason for the decrease in the SRPC’s strength in the later stage. With the increase in the curing temperature and age, the number of harmful pores in the SRPC increases, especially during curing at 60 °C. Pores of approximately 1000 nm in size can be found in the hardened paste, and the microstructural stability of the SRPC decreases, resulting in a decrease in the macroscopic strength [28]. Table 4 and Table 5 show a reliability analysis of the compressive strength and flexural strength test results, which are in general agreement with the test conclusions.

From Figure 2, it can be found that when the temperature is less than 50 °C, the strength development of the SRPC and the OPC is similar under the same curing conditions, but the 90-day strength of the OPC exhibits no obvious downward trend. At 60 °C, the compressive strength of the OPC decreases sharply after 28 d, while that of the SRPC decreases only slightly at 90 d, which indicates that the high-temperature resistance of the SRPC is better than that of the OPC. The development law of the flexural strength of both cement types is basically the same when cured at high temperatures, but the flexural strength of SRPC is higher under the same curing conditions.

### 3.2. Pore Structure

Pore structure is not only the decisive factor affecting the durability of hardened cement pastes but is also one of the key factors affecting the strength of cement-based materials [29]. The effects of different sizes of holes on a cement’s strength can be divided into the following four categories [30]: harmless holes (<20 nm), less harmful holes (20~50 nm), harmful holes (50~200 nm), and multiple harmful holes (>200 nm). The pore structure and pore size distribution of the SRPC at different curing ages and different curing temperatures are shown in Figure 3 and Figure 4.

Figure 3 shows the development law of the mortar’s pore structure with age at 60 °C. With the increase in the curing age, the internal porosity of the SRPC gradually decreases. The most probable pore size is approximately 350 nm at 1 d. With the formation of hydration products, some pores are filled, and the most probable pore size decreases to 227 nm, 95 nm, and 32 nm at 7 d, 28 d, and 90 d, respectively. However, the high temperatures had an adverse effect on the development of the pore structure in the later stage of hydration. Although most of the pores in the hardened cement paste were less than 50 nm in diameter after 90 d, there were also some harmful pores whose pore diameters were approximately 1000 nm. The reason for this is that the hydration products were unevenly dispersed due to the accelerated hydration at high temperatures, so the pores in the paste could not be filled completely and the compactness of the paste was reduced [6]. This is also the reason why the growth of the SRPC’s strength was restricted in the later stage.

Figure 4 shows the mortar’s pore structure at different curing temperatures. It can be seen that the number of harmful holes in the SRPC increases gradually with the increase in the curing temperature. At 20 °C, the size of the pores in the paste is mainly in the range of 10~50 nm, and the peak value is about 40 nm, indicating that there are more pores at the pore level. According to the results shown in Table 6, it can also be seen that pores with a size of less than 50 nm account for 74.3% of the total number of pores. The pore structure distribution at 40 °C is similar to that at 20 °C, but the number of pores whose size is greater than 200 nm is 8% higher than that at 20 °C. When the curing temperature reached 50 °C, the pore structure of the SRPC coarsened, the pore size distribution curve moved in the direction of large pore diameters, and the number of harmful pores (50~200 nm in size) increased sharply, accounting for 46.5% of the total number of pores (28.6% higher than that at 20 °C). At 60 °C, the pore structure further coarsened, the proportion of multiple harmful pores (pores larger than 200 nm in size) increased by 17.9% compared with 20 °C, and most of the pores were approximately 1000 nm in size.

### 3.3. Microscopic Analysis

#### 3.3.1. Hydration Products

Figure 5 shows the XRD patterns of the SRPC at different temperatures (Figure 5a) and different curing ages (Figure 5b). It can be seen from Figure 5a that, compared with the XRD pattern of OPC [9], high-temperature curing had little effect on the types of hydration products of SRPC. Yan et al. [31] also found that high temperatures hardly affect the formation types of hydration products. However, it can be found from Figure 5 that the diffraction peak of ettringite gradually weakened as the curing temperature increased. In Figure 5b, a weak diffraction peak of ettringite can be observed at 1 day only and, due to the progression of hydration, only Ca(OH)_2_ and hydrotalcite can be observed at 28 d.

The composition of the cementitious system for the hydration of cement is relatively complex. Even ordinary Portland cement contains at least seven different hydration products, most of which are solid solutions with a variable composition [17]. It is difficult to fully show the state change of hydration products by an XRD test. The hydration reaction of cement can be simulated by GEMS, and the change in the hydration products can be better understood by combining the simulation results with specific XRD analysis results [32]. Based on this, the hydration degree of the SRPC with time was simulated by GEMS, and the result is shown in Figure 6. It can be seen from Figure 6 that the types of hydration products did not change with the increase in the curing temperature, but the hydration process of the cement was accelerated. Compared with 20 °C, in the temperature range of 40~60 °C, the consumption of calcium silicate (C_3_S) and other substances increased significantly, and the production of C-S-H and other hydration products increased. Taking 3 days of hydration as the time node, the degree of dissolution of C_3_S at 20 °C was 67.1% and it increased to 78.7%, 83.1%, and 86.2% at 40~60 °C, respectively. The production of C-S-H also increased from 72.7% to 82.5%, 86.2%, and 89.3%. Moreover, Figure 6 shows that the reaction amount of C_3_S at 40~60 °C for 7 d is 1.1 times, 1.14 times, and 1.2 times that at room temperature, which also proves that the strength of the cement increased faster under high-temperature curing conditions. In addition, it can be seen from Figure 2a that at 20 °C, the compressive strength of the SRPC at 7 d is 32.8 MPa, which is 62.7% of the 28-day strength (52.3 MPa). Combined with the predicted results shown in Figure 6a, these results show that the hydration degree of the cementitious system at 7 d is approximately 60% of that at 28 d, and the strength development is in good agreement with the results of the simulation of the hydration process.

Figure 7 shows the change in the hydration products with the increase in the curing temperature. It can be seen that the main hydration products C-S-H, Ca(OH)_2_, and hydrotalcite are relatively stable at 60 °C, and hydrogarnet also had no material transformation in this temperature range. However, when the temperature increased to 50 °C, the monocarbonate disappeared, the amount of ettringite decreased (2.23%), and the amount of monosulfate (AFm) increased (3.06%). This was because, at this temperature, ettringite reacts with monocarbonate to form AFm [33]. AFm can be formed at 34 °C, and the amount increases as the temperature increases and the hydration reaction continues. Perkins et al. [34] found that increasing the temperature promoted the formation of AFm and decreased the stability of ettringite. In addition to material transformation, high-temperature dehydration also affects the stability of ettringite [9]. When the temperature exceeds 50 °C, ettringite crystals will lose two molecules of bound water, the crystallinity will decrease [35], and the diffraction intensity will also decrease in the XRD pattern. This is why the diffraction peak of ettringite cannot be observed in Figure 4.

Since the amount of tricalcium aluminate (C_3_A) in the SRPC is small, and the hydration rate of C_3_A is high, it is difficult to quantitatively analyze it in the overall hydration process [32]. Therefore, the hydration process of C_3_A was studied separately, and the results are shown in Figure 7b. The hydration rate of C_3_A mainly depends on the temperature (the higher the temperature, the higher the hydration rate) [36]. It can be seen that the hydration degree of C_3_A in the SRPC under high-temperature curing conditions is higher than that under normal-temperature curing conditions. The C_3_A can react completely within 24 h, which indicates that it will soon be consumed in the hydration reaction at high temperatures, and the amount of residual alumina after hydration will also be smaller. Moreover, the ettringite will undergo material transformation at high temperatures, which increases the consumption of tricalcium aluminate hydrate in the cementitious system. The reduction in the amount of aluminum phase means that when sulfate corrosion is encountered in a service process, the amounts of substances that can participate in the reaction are reduced, and the corrosion reaction is slowed down [5].

Moreover, the material transformation also affects the total amount of hardened paste. The density of ettringite is lower and its molecular volume is higher. The consumption of ettringite at high temperatures leads to a decrease in the total amount of hardened paste, an increase in the internal porosity [34], and a decrease in the microstructure’s stability. So, the strength of SRPC decreases as the curing temperature increases.

#### 3.3.2. Microstructure

Figure 8 shows the microstructure of the SRPC hydrated for 1 d, 3 d, 7 d, and 28 d at 60 °C. It can be seen that the internal compactness of the hardened cement paste is affected to a certain extent. The hydration products that were formed could not be dispersed due to the influence of the newly formed products before they could be filled into the voids, and hydration products are wrapped around the cement clinker. As a result, the hydration process did not complete, resulting in an uneven distribution of the hydration products in the hardened cement paste.

Figure 9 shows the microstructure of the SRPC at a curing temperature of 20~60 °C after 28 d. It can be seen that the distribution of hydration products is relatively uniform at 20 °C, and the degree of compactness of the hardened paste is higher. However, the internal compactness of the paste decreases at 40~60 °C. The morphology of the AFt is fine and needle-like at 20 °C and 40 °C. With the increase in the curing temperature, the hydration rate increases, the amount of hydration products increases, the crystal grows continuously, and the morphology of the AFt becomes thicker and irregularly rod-like at 50 °C, resulting in a larger amount of expansion stress inside the cement paste [37]. As the stability of the AFt decreases at 50 °C, the material transformation begins to form AFm. When the temperature increases to 60 °C, as more and more AFt transforms, the amount of material with a crystal morphology decreases, and the amount of petal-like AFm increases. Yuan et al. [38] also found that the morphology of AFt changes with an increase in the temperature, which is also one of the reasons for the increase in the number of pores in the hardened paste at high temperatures.

## 4. Conclusions

In this study, the strength development, pore structure, hydration products, and micro-morphology of SRPC under high-temperature curing conditions were analyzed. Our main conclusions are as follows:

High-temperature curing promotes an increase in the early strength of SRPC, but, at the same time, it also limits the development of the SRPC’s strength in the later stage. Compared with normal-temperature curing, the compressive strength of the SRPC at 40~60 °C was increased by 38.2~54.8% at 3 d. The compressive strength of the SRPC at all curing temperatures was basically the same at 28 d, but the compressive strength at 90 d was 3.4~13.5% lower than that at the normal temperature. The flexural strength of the SRPC under high-temperature curing was also higher than that under normal-temperature curing at the early stage, but the flexural strength decreased by 7.1~23.3% at 90 d;The internal pore diameter of the SRPC coarsened under high-temperature curing, which limits the growth of strength in the later stage. Compared with normal-temperature curing, the number of harmful pores (50~200 nm in size) inside the hardened paste increased continuously at curing temperatures of 40 °C and 50 °C. The number of multiple harmful holes (>200 nm in size) increased by 17.9% at 60 °C. The longer the curing time at high temperatures, the larger the number of harmful holes in the SRPC, which has a greater impact on the growth of the strength of the SRPC;The types of hydration products of SRPC did not obviously change under high-temperature curing, but the ettringite changed with the increase in the temperature. There was no obvious change in the types of hydration products at 40 °C. When the temperature exceeded 50 °C, ettringite reacted with monocarbonate to form monosulfate, resulting in changes in the microstructure of the paste. The increase in the curing temperature accelerated the hydration process of the cement and the reaction rate of the early cement hydration process increased. The higher the temperature, the faster the reaction rate. The amount of hydration products at 60 °C for 7 d was 1.56 times higher than that at 20 °C. During the hydration of SRPC under high-temperature curing, more tricalcium aluminate hydrate is consumed to form AFm, which can inhibit the formation of “delayed ettringite” during the SRPC’s service life and improve its ability to resist sulfate erosion.

## Figures and Tables

**Figure 1 materials-15-02790-f001:**
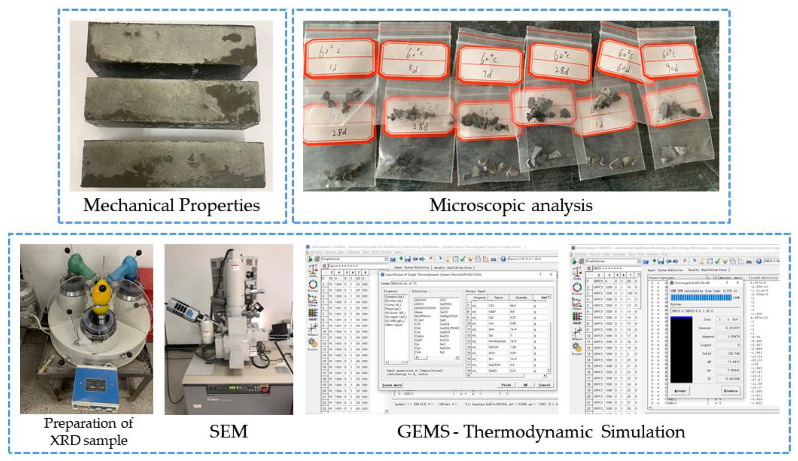
Preparation and testing stages.

**Figure 2 materials-15-02790-f002:**
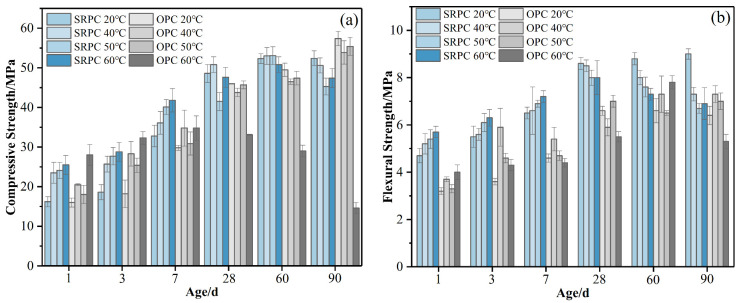
Variation in the specimens’ strength with age under different hydration temperatures: (**a**) compressive strength; (**b**) flexural strength.

**Figure 3 materials-15-02790-f003:**
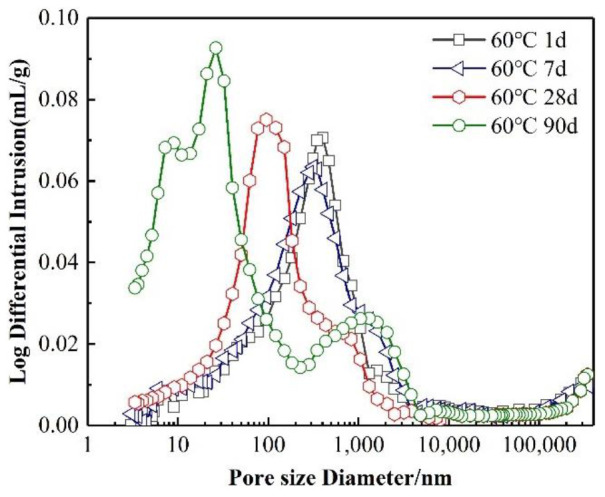
Pore structure of the SRPC at different ages.

**Figure 4 materials-15-02790-f004:**
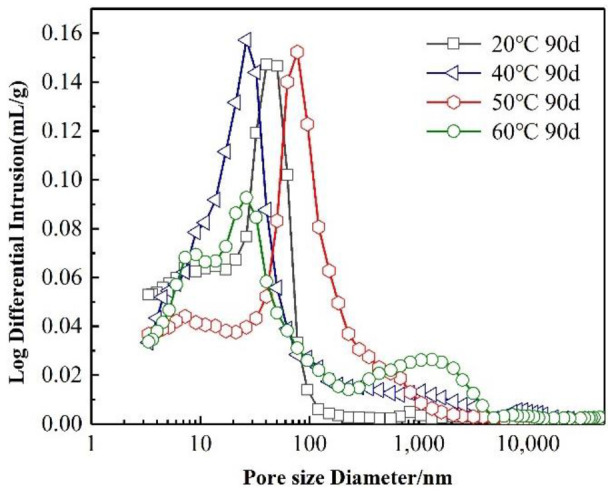
Pore structure of the SRPC at different temperatures.

**Figure 5 materials-15-02790-f005:**
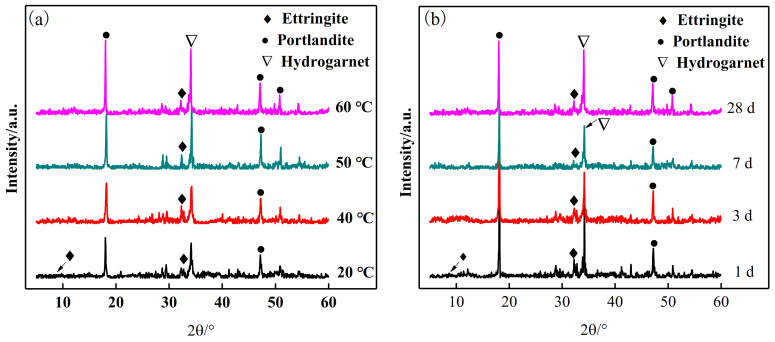
XRD patterns of SRPC hydration products: (**a**) at different temperatures; (**b**) at different ages.

**Figure 6 materials-15-02790-f006:**
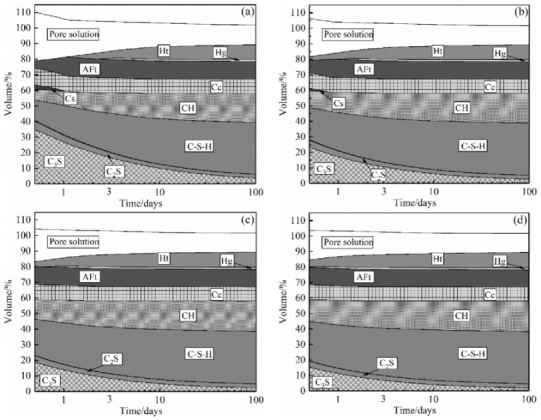
Relationship between hydration products and time: (**a**) 20 °C; (**b**) 40 °C; (**c**) 50 °C; (**d**) 60 °C. C_3_S, Tricalcium Silicate; C_2_S, Dicalcium Silicate; C-S-H, Calcium Silicate Hydrate; CH, Calcium Hydroxide; Cs, Gypsum; Cc, Calcite; AFt, Ettringite; Ht, Hydrotalcite; Hg, Hydrogarnet.

**Figure 7 materials-15-02790-f007:**
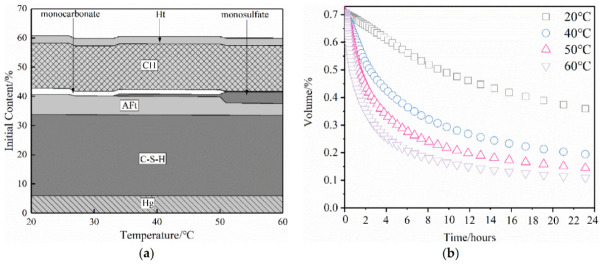
Relationship between hydration products and temperature: (**a**) change in hydration products with temperature; (**b**) hydration of C_3_A at different temperatures. C-S-H, Calcium Silicate Hydrate; CH, Calcium Hydroxide; AFt, Ettringite; Ht, Hydrotalcite; Hg, Hydrogarnet.

**Figure 8 materials-15-02790-f008:**
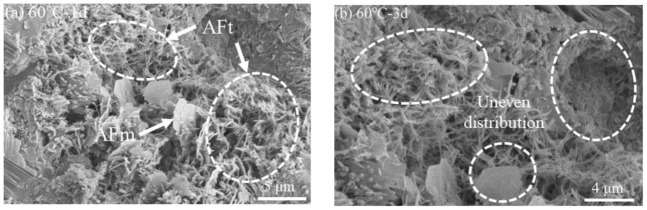
Microstructure of the SRPC at a curing temperature of 60 °C: (**a**) 1 d; (**b**) 3 d; (**c**) 7 d; (**d**) 28 d.

**Figure 9 materials-15-02790-f009:**
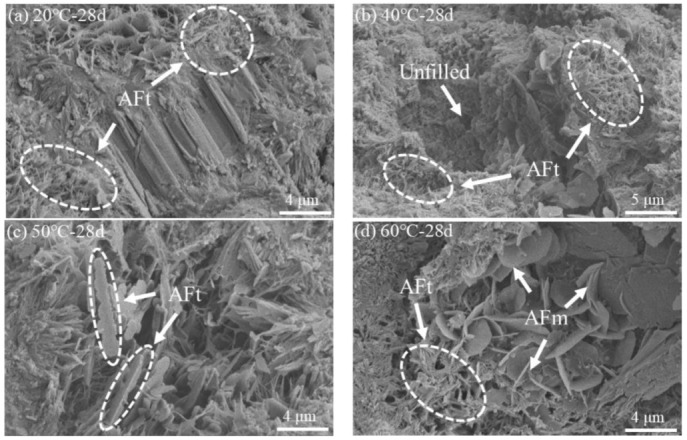
Microstructure of the SRPC at different curing temperatures: (**a**) 20 °C; (**b**) 40 °C; (**c**) 50 °C; (**d**) 60 °C.

**Table 1 materials-15-02790-t001:** Chemical composition of the cement (%).

Cement	CaO	SiO_2_	Al_2_O_3_	Fe_2_O_3_	SO_3_	MgO	K_2_O	Na_2_O	P_2_O_5_	LOI
SRPC	59.5	17.6	1.9	8.5	3.2	1.7	0.8	0.2	0.1	6.4
OPC	61.83	19.68	5.72	5.66	1.83	1.31	1.27	0.43	0.8	2.23

**Table 2 materials-15-02790-t002:** Physical properties of the cement.

Cement	Density (g/cm^3^)	Initial Setting Time/Min	Final Setting Time/Min	Specific Surface Area (m^2^/kg)	Flexural Strength/MPa		Compressive Strength/MPa	
					3 d	28 d	3 d	28 d
SRPC	3.19	150	275	299	5.4	8.9	27.9	49.4
OPC	2.82	135	225	325	3.9	7.1	20.9	45.6

**Table 3 materials-15-02790-t003:** Early strength growth rate of the SRPC under high-temperature curing (relative to 20 °C).

Age/Days	Increasing Rate of Compressive Strength/%			Increasing Rate of Flexural Strength/%		
	40 °C	50 °C	60 °C	40 °C	50 °C	60 °C
1	45.1	48.8	57.4	10.6	14.9	21.3
3	38.2	48.9	54.8	1.8	10.9	14.5
7	10.1	22.3	27.4	1.8	6.2	10.8
28	4.5	−14.6	−2.1	−1.1	−9.1	−18.8
60	1.3	−1.5	−2.8	−6.9	−13.6	−25.5
90	−3.4	−13.5	−9.5	−7.1	−16.9	−23.3

**Table 4 materials-15-02790-t004:** Reliability analysis of compressive strength test results.

Text/Mixture	Average 1/3/7/28/60/90 d *f*_c_ (MPa)	St. Dev. (MPa)	*p*-Value
20 °C	16.2/18.6/32.8/48.6/52.3/52.4	1.2/1.9/2.6/2.2/1.2/1.8	-
40 °C	23.5/25.7/36.1/50.8/53.0/50.6	1.9/2.0/2.8/1.9/2.0/1.8	0.022/0.0885/0.1037/ 0.4429/0.8149/0.0059
50 °C	24.1/27.7/40.1/41.5/53.1/45.3	2.3/2.1/1.9/2.3/2.4/2.1	0.024/0.0635/0.0826/ 0.0019/0.7900/0.0060
60 °C	25.5/28.8/41.8/47.6/50.8/47.4	1.1/2.3/2.8/2.5/1.9/2.4	0.002/0.0008/0.0579/ 0.7922/0.1674/0.0219

**Table 5 materials-15-02790-t005:** Reliability analysis of flexural strength test results.

Text/Mixture	Average 1/3/7/28/60/90 d *f*_c_ (MPa)	St. Dev. (MPa)	*p*-Value
20 °C	4.7/5.6/6.5/8.6/8.58/9.0	0.30/0.40/0.26/0.26/0.27/0.21	-
40 °C	5.2/5.6/6.6/8.5/8.0/7.3	0.40/0.20/0.95//0.23/0.31/0.30	0.383/0.789/0.874/ 0.383/0.070/0.008
50 °C	5.4//6.1/6.9/8.0/7.6/6.7	0.30/0.40/0.20/0.29/0.43/0.22	0.205/0.102/0.253/ 0.009/0.065/0.003
60 °C	5.7/6.3/7.2/8.0/7.3/3.9	0.20/0.30/0.26/0.73/0.25/0.69	0.082/0.015/0.136/ 0.2170.005/0.021

**Table 6 materials-15-02790-t006:** Aperture ratio of the SRPC.

Age/Days	Aperture Range/nm				Temperature/℃	Aperture Range/nm			
	<20	20–50	50–200	>200		<20	20–50	50–200	>200
1	6.1%	7.3%	24.1%	62.5%	20	39.6%	34.7%	17.9%	7.8%
7	7.2%	7.2%	27.3%	58.3%	40	40.7%	31.7%	11.8%	15.8%
28	27.5%	13.8%	39.9%	18.8%	50	23.5%	13.1%	46.5%	16.8%
90	38.3%	22.7%	13.3%	25.7%	60	38.3%	22.7%	13.3%	25.7%

## Data Availability

Authors have checked the data and comfirmed that no ethics issue exist in this paper.

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
