# Peer review of "Study on the Strength and Hydration Behavior of Sulfate-Resistant Cement in High Geothermal Environment"

_materials, 2022, doi:10.3390/ma15082790_

Round 1
Reviewer 1 Report
Effect of high temperature curing on the strength development, pore structure, hydration products and micromorphology of Sulphate Resistant Portland cement (SRPC) has been specified. To do it, thermodynamic simulation based on Gibbs free energy minimization program (GEMS) was applied.
All in all, the manuscript has an acceptable situation. After removing all the following comments, the publication of the manuscript is feasible.
1. What is the GEMS abbreviation proposed in the abstract section? Each abbreviation in the manuscript at the first appearance needs to be mentioned as a completed phrase.
2. Please consider a nomenclature list and define all the applied parameters,
variables and abbreviations appeared in the manuscript.
3. Please review those papers published in the recent year, 2022, and add them to the introduction section.
4. Consideration of a schematic (or realistic) figure for preparation and testing stages in subsections 2.2 and 2.3 could be interesting. Please add it.
5. Referring to the 'Porous medium' and the other related keywords in the text of the manuscript is necessary.
6. Proposed novelty of the manuscript against the prior valid works in the
literature is not clear. Please determine it, exactly.
7. Please consider a logical space between tables 1 and 2.
8. Caption of the multi- figures is not suitable. Descriptions around figures (a), (b), and so on should be added in a common caption, independently.
Author Response
Dear Professor:
Thank you very much for your attention and the referee¢s evaluation and comments on our manuscript “Study on Strength and Hydration Mechanism of Sulfate Resistant Cement under High Geothermal Environment”. We have revised the manuscript according to your kind advices and referee¢s detailed suggestions. Enclosed please find the responses to the referees. We sincerely hope this manuscript will be finally acceptable to be published on Materials. Thank you very much for all your help and looking forward to hearing from you soon. The revised parts of the article are marked in blue font.
Response to the referee¢s comments
I am very grateful to your comments for the manuscript. Thank you for your suggestions, those suggestions are very important, they have important guiding significance for my writing and research work. According with your advice, we amended the relevant part in manuscript. Some of your questions are answered below.
Effect of high temperature curing on the strength development, pore structure, hydration products and micromorphology of Sulphate Resistant Portland cement (SRPC) has been specified. To do it, thermodynamic simulation based on Gibbs free energy minimization program (GEMS) was applied.
All in all, the manuscript has an acceptable situation. After removing all the following comments, the publication of the manuscript is feasible.
(1) What is the GEMS abbreviation proposed in the abstract section? Each abbreviation in the manuscript at the first appearance needs to be mentioned as a completed phrase.
Reply:
The specific name of GEMS has been described in detail in the abstract of the manuscript, and the abbreviation nouns first appeared in the manuscript are also described in detail.
(2) Please consider a nomenclature list and define all the applied parameters, variables and abbreviations appeared in the manuscript.
Reply:
We summarize the parameters, variables and abbreviations mentioned in the full text, and give a nomenclature list, as shown in the table at the end of the article.
(3) Please review those papers published in the recent year, 2022, and add them to the introduction section.
Reply:
We have reviewed the literature published in recent years, and have added it to the introduction section of the article.
(4) Consideration of a schematic (or realistic) figure for preparation and testing stages in subsections 2.2 and 2.3 could be interesting. Please add it.
Reply:
In Section 2.3 of the manuscript, we added actual pictures of the experiment during the preparation and operation stages, including mechanical and microscopic tests.
(5) Referring to the “Porous medium” and the other related keywords in the text of the manuscript is necessary.
Reply:
Thank you very much for your suggestion, and we also found that the test results show that the porosity of SRPC is a point worthy of attention. Therefore, we follow your opinion and add “porous” to the keywords of the manuscript.
(6) Proposed novelty of the manuscript against the prior valid works in the literature is not clear. Please determine it, exactly.
Reply:
We apologize for the lack of sufficient description of the novelty of the content of the manuscript studied in our initial submission. In this revision, we summarize the limitations of the previous study, and then elaborate the novelty of this study in the first sentence of the last paragraph of the introduction.
(7) Please consider a logical space between tables 1 and 2.
Reply:
Thank you very much for your suggestion. There are indeed problems in the logical relationship between Table 1 and Table 2, and we have unified the logical relationship between Table 1 and Table 2 in the revised version.
(8)Caption of the multi- figures is not suitable. Descriptions around figures (a), (b),
and so on should be added in a common caption, independently.
Reply:
We apologize for the fact that when we first submitted the manuscript, we only
marked it inside the picture when there were multiple pictures in the manuscript, and
did not describe it in detail in the title. In this revision, we modify the places where
multiple pictures appear, and describe the meaning represented by each picture in the
picture name.

Reviewer 2 Report
The manuscript Study on Strength and Hydration Mechanism of Sulfate Resistant Cement under High Geothermal Environment submitted to Materials - Manuscript Number: materials-1657076 describes the preparation and characterization of cement mortar specimens at different curing temperatures including the evolution of microscopic morphology and pore structure of SRPC.
Some remarks have to be taken into account by the authors:
- Sufficient details of materials should be described to allow others to replicate and build on published results. The lacks of specification and source/origin for the used materials, chemicals and reagents. Please add to the main text.
- How many samples of one type (or the number of replicates of the experiment) were used in the studies? Lack of information about the number of replicates of the experiment.
- How is the reproducibility of the used procedures? This should be evidenced by the repeating of the preparation experiments and characterization and evaluation of the results.
- Error range or the determination coefficient should be included e.g. Tables. Statistical analysis should be emphasized to support the results, discussion and conclusion. Please correct.
Author Response
Dear Professor:
Thank you very much for your attention and the referee¢s evaluation and comments on our manuscript “Study on Strength and Hydration Mechanism of Sulfate Resistant Cement under High Geothermal Environment”. We have revised the manuscript according to your kind advices and referee¢s detailed suggestions. Enclosed please find the responses to the referees. We sincerely hope this manuscript will be finally acceptable to be published on Materials. Thank you very much for all your help and looking forward to hearing from you soon. The revised parts of the article are marked in blue font.
Response to the referee¢s comments
I am very grateful to your comments for the manuscript. Thank you for your suggestions, those suggestions are very important, they have important guiding significance for my writing and research work. According with your advice, we amended the relevant part in manuscript. Some of your questions are answered below.
The manuscript Study on Strength and Hydration Mechanism of Sulfate Resistant Cement under High Geothermal Environment submitted to Materials - Manuscript Number: materials-1657076 describes the preparation and characterization of cement mortar specimens at different curing temperatures including the evolution of microscopic morphology and pore structure of SRPC.
Some remarks have to be taken into account by the authors:
- Sufficient details of materials should be described to allow others to replicate and build on published results. The lacks of specification and source/origin for the used materials, chemicals and reagents. Please add to the main text.
Reply:
The material used in the manuscript is mainly different from the source, we have given the detailed information of the test material production company in section 2.1 of the manuscript.
- How many samples of one type (or the number of replicates of the experiment) were used in the studies? Lack of information about the number of replicates of the
Reply:
The test is carried out with reference to the Chinese standard “Method of testing cements-Determination of strength, GB/T 17671-1999”, which is formulated by Chinese authorities to ensure the standardization of the test. All materials are prepared and tested in accordance with the requirements of the standard. At the same time, there are 72 groups of samples in the mechanical test and 36 groups in the microscopic test, and the reliability of each test data meets the requirements of the standard.
- How is the reproducibility of the used procedures? This should be evidenced by the repeating of the preparation experiments and characterization and evaluation of the
Reply:
The test was carried out in full accordance with the test criteria and the data is highly reliable. The procedure used in the test has been recognized by many researchers in the field and has been widely used in the study of cementitious materials. The research method of GEMS simulating cement hydration has been established, such as the research results obtained in References 17, 19 and 20 in the manuscript. Therefore, the method was used in this test to study the hydration behaviour of SRPC and the test results are indicative.
- Error range or the determination coefficient should be included e.g. Tables. Statistical analysis should be emphasized to support the results, discussion and conclusion. Please correct.
Reply:
In view of the error of the test data, we add the error bar to figure 2 of the manuscript, and analyze the reliability of all the data in the figure, the results are shown in Table 4 and Table 5, and the analysis results are in good agreement with the test conclusions.

Round 2
Reviewer 1 Report
The authors answered the comments precisely and thus the paper can be accepted in the current form.